

**Consistency evaluation of tropospheric ozone from ozonesonde and**
**IAGOS aircraft observations: vertical distribution, ozonesonde types**
**and station-airport distance**
Honglei Wang[1, 2], David W. Tarasick[3*], Jane Liu[2*], Herman G.J. Smit[4], Roeland Van Malderen[5],
Lijuan Shen[6], Tianliang Zhao[1]
[1] China Meteorological Administration Aerosol-Cloud and Precipitation Key Laboratory, Nanjing University of
Information Science and Technology, Nanjing 210044, China;
[2] Department of Geography and Planning, University of Toronto, Canada;
[3] Environment and Climate Change Canada, 4905 Dufferin Street, Downsview, ON, M3H 5T4 Canada;
[4] Institute for Energy and Climate Research: Troposphere (IEK-8), Research Centre Juelich (FZJ), Juelich, Germany;
[5] Royal Meteorological Institute of Belgium, Brussels, Belgium;
[6] School of Atmosphere and Remote Sensing, Wuxi University, Wuxi 214105, China.
* Corresponding authors: David.Tarasick@ec.gc.ca and janejj.liu@utoronto.ca
**Abstract:** The vertical distribution of tropospheric $O_3$ from ozonesondes is compared with that from
In-service Aircraft for a Global Observing System (IAGOS) measurements at 23 pairs of sites
between about 30°S and 55°N, from 1995 to 2021. Profiles of tropospheric $O_3$ from IAGOS aircraft
are in generally good agreement with ozonesonde observations, for Electrochemical concentration
cells (ECC), Brewer-Mast, and Carbon-Iodine sensors, with average biases of 7.03 ppb, 6.28 ppb,
and -4.48 ppb, and correlation coefficients (R) of 0.72, 0.86, and 0.82, respectively. Agreement
between the aircraft and Indian-sonde observations is poor, with an average bias of 24.07 ppb and
R of 0.41. The $O_3$ concentration observed by ECC sondes is on average higher by 5-10% than that
observed by IAGOS aircraft, and the relative bias increases modestly with altitude. For other sonde
types, there are some seasonal and altitude variations in the relative bias with respect to IAGOS
measurements, but these appear to be caused by local differences.
The distance between station and airport within 4° has little effect on the comparison results. For
the ECC ozonesonde, the overall bias with respect to IAGOS measurements varies from 5.7 to 9.8
ppb, when the station pairs are grouped by station-airport distances of <1° (latitude and longitude),
1-2°, and 2-4°. Correlations for these groups are R = 0.8, 0.9 and 0.7. These comparison results



provide important information for merging ozonesonde and IAGOS measurement datasets. They
can also be used to evaluate the relative biases of the different sonde types in the troposphere, using
the aircraft as a transfer standard.
**Key words:** WOUDC; IAGOS; tropospheric $O_3$; vertical distribution; ozonesonde; aircraft

**1 Introduction**
Ozone ($O_3$) is a trace gas with small concentrations in the atmosphere (Ramanathan et al., 1985); it
is an important greenhouse gas in the atmosphere. In the planetary boundary layer, it is a major air
pollutant (Lefohn et al., 2018; Monks et al., 2015). It can endanger human health, damage
ecosystems, and affect climate change (Fu and Tai, 2015; Lefohn et al., 2018; Percy et al., 2003).
Therefore, it is of importance to study the temporal and spatial distribution and the factors and
mechanisms affecting the variation of tropospheric $O_3$ including near-surface $O_3$ (Logan, 1985; Ma
et al., 2020; Sharma et al., 2017; Young et al., 2018).
A large number of studies have been carried out on the spatiotemporal distribution, formation
mechanism, and transport characteristics of ground $O_3$ (Li et al., 2020, 2021; Vingarzan, 2004; Wang
et al., 2017, 2023; Xu et al., 2021; Yu et al., 2021). However, due to the limitation of observations,
there are many unknowns on tropospheric ozone, especially the vertical distribution of tropospheric
$O_3$. Satellites provide an effective platform for measuring $O_3$ globally. Satellite $O_3$ instruments,
including TES, GOME, GOME-2, SCIAMACHY, OMI, and TROPOMI, have been in operation for
decades (David et al., 2013; Ebojie et al., 2016; Hegarty et al., 2009; Hoogen et al., 1999; Hubert et
al., 2021; Miles et al., 2015). Although satellite observations can provide detailed temporally- and
horizontally-resolved maps of tropospheric $O_3$ columns, in general satellite data lack vertical
resolution. While tropospheric differential absorption lidar can also provide vertical distribution
information for tropospheric $O_3$ (Keckhut et al., 2004; Yang et al., 2023), there are very few routinely
operating stations.
The principal sources of vertically-resolved, trend-quality observations of tropospheric $O_3$ are
therefore balloon-borne ozonesondes, and IAGOS aircraft observations. The World Ozone and
Ultraviolet Radiation Data Centre (WOUDC) and the In-service Aircraft for a Global Observing
System database (IAGOS) house the data from these two observation programs with the longest

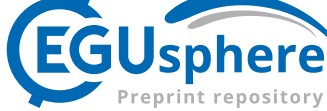

duration and the most global stations, which are the most widely used for tropospheric $O_3$ studies
(Gaudel et al., 2020; Liao et al., 2021; Tarasick et al., 2019; Wang et al., 2022). These two datasets
are used to study the distribution, variability and trends of tropospheric $O_3$, and its sources and
transport, as well as satellite and model validation (Hu et al., 2017; Gaudel et al., 2018; 2020; Wang
et al., 2022; Zhang et al., 2008). The first phase of the Tropospheric Ozone Assessment Report
(TOAR-I), initiated in 2014, utilized available surface, ozonesonde, aircraft, and satellite
observations to assess tropospheric $O_3$ trends from 1970 to 2014 (Schultz et al., 2017). Hu et al.
(2017) found that the largest bias in a chemical transport model, GEOS-Chem, with respect to
ozonesondes and IAGOS observations, is in high northern latitudes in winter-spring, where the
simulated ozone is 10-20 ppb lower. Wang et al. (2022) examined observed tropospheric $O_3$ trends,
their attributions, and radiative impacts from 1995 to 2017, using aircraft observations from IAGOS,
ozonesondes, and a multi-decadal GEOS-Chem chemical model simulation, and found increases in
tropospheric ozone (950 - 250 hPa) of $2.7 \pm 1.7$ ppbv per decade from IAGOS observations in the
Northern Hemisphere and at 19 of 27 global ozonesonde sites averaging $1.9 \pm 1.7$ ppbv per decade.
There are also a number of comparative studies on these two datasets (Zbinden et al., 2013; Staufer
et al., 2013, 2014; Tanimoto et al., 2015; Tarasick et al., 2019). Staufer et al. (2013, 2014) used
trajectory calculations to match air parcels sampled by both sondes and aircraft. Zbinden et al. (2013)
compared coincidences (±24 hours) at three site pairs, while Tanimoto et al. (2015) examined
simultaneous observations (±3 hours for sonde versus aircraft) at several site pairs less than 100 km
apart. In general, these studies show small (6% or less) negative biases of aircraft measurements
against ECC sondes. Tarasick et al. (2019) compared trajectory-mapped averages over 20-70 N of
ozonesonde and MOZAIC/IAGOS profiles and concluded that over 1994-2012 ozonesonde
measurements were about $5 \pm 1\%$ higher in the lower troposphere and $8 \pm 1\%$ higher in the upper
troposphere.
In this study, we attempt to make the most comprehensive evaluation to date of the relative biases
of IAGOS and sonde profiles, using as many station pairs as possible. We identify 23 suitable pairs
of sites in the WOUDC and IAGOS datasets from 1995 to 2021, compare the average vertical
distribution of tropospheric $O_3$ shown by ozonesonde and aircraft measurements, and analyze their
differences by ozonesonde type and by station-airport distance.




## 2 Data and methods

### 2.1 MOZAIC-IAGOS observations

The MOZAIC (Measurements of OZone and water vapor on Airbus In-service airCraft) program, initiated in 1994 and incorporated into the IAGOS (In-service Aircraft for a Global Observing System; www.iagos.org) program since 2011, takes advantage of commercial aircraft to provide worldwide in-situ measurements of several trace gases (e.g., $O_3$ and CO) and meteorological variables (e.g., water vapor) throughout the troposphere and the lower stratosphere (Marenco et al., 1998; Petzold et al., 2015; Nédélec et al., 2015). $O_3$ measurements are performed using a dual-beam UV-absorption monitor (time resolution of 4 seconds) with an instrumental uncertainty of ±2 ppbv+2% (Thouret et al., 1998; Blot et al., 2021). It should be noted that this is only the instrumental uncertainty, and does not include sampling uncertainties (possible losses) caused by the inlet line and the compressor before the UV-photometric measurements are made. Loss of ozone on the inlet pump was an issue in earlier aircraft ozone sampling programs (Brunner et al., 2001; Dias-Lalcaca et al., 1998; Schnadt Poberaj et al., 2007), but Thouret et al. (1998) found it negligible for MOZAIC/IAGOS.

More details on the new IAGOS instrumentation can be found in Nédélec et al. (2015). The continuity of the dataset between the MOZAIC and IAGOS programs has been demonstrated based on their 2-year overlap (2011~2012) (Nédélec et al., 2015). Blot et al. (2021) performed an evaluation of the internal consistency of the $O_3$ and CO measurements since 1994, which confirmed the instrumental uncertainty of ±2 ppbv. Moreover they found no drift in the bias amongst the different instrument units (six $O_3$ and six CO IAGOS-MOZAIC instruments, nine IAGOS-Core Package1 and the two instruments used in the IAGOS-CARIBIC aircraft).

### 2.2 WOUDC ozonesonde observations

The World Ozone and Ultraviolet Radiation Data Centre (WOUDC) is part of the Global Atmosphere Watch (GAW) program of the World Meteorological Organization (https://woudc.org/data/explore.php). The WOUDC is operated by Environment and Climate Change Canada. WOUDC ozonesonde data have been evaluated in a number of WMO-sponsored international field intercomparisons (Attmannspacher and Dütsch, 1970, 1981; Kerr et al, 1994) and



more recently in laboratory simulation chamber experiments using a standard reference photometer
(Smit et al., 2007, 2024; Thompson et al., 2019). In the global ozonesonde network, while different
ozonesonde types were common in the past, more than 95% of current sounding stations use
electrochemical concentration cells (ECC). ECC ozonesondes have a precision of 3-5% (1-σ) while
the precision of other sonde types is somewhat poorer, at about 5–10% for Brewer-Mast and the
Japanese KC (Carbon-Iodine) sonde, and somewhat larger for the Indian-sonde (Kerr et al., 1994;
Smit et al., 2007). Biases with respect to UV reference spectrometers have been estimated for ECC
sondes at 1-5% in the troposphere (Smit et al., 2021; Tarasick et al., 2019, 2021).
**2.3 Data processing**
The two datasets were first screened for airport-sonde station pairs within a latitude separation of
<4° and a longitude separation of <4°. Many sonde stations have observational records that do not
overlap with the IAGOS period (1994-present). In addition, the IAGOS dataset has large gaps at
many airports, because the frequency of visits to airports by aircraft that take part in IAGOS depends
on commercial airlines' operating constraints. In total, 23 station pairs (Fig. 1) were identified with
a separation of less than 4° in both latitude and longitude, and coincident observations over at least
nine months. The majority of the 23 ozonesonde site records are ECC (17), while four are Indian-
sonde, one Brewer-Mast, and one Carbon-Iodine (the Japanese KC sonde). These stations were
divided into 3 groups according to the distance (D) between the ozonesonde station and the airport:
D<1°, 1°<D<2°, and 2°<D<4°. Specific information on the comparison stations is shown in Table

1.

The observation times of the ozonesonde and aircraft are generally not the same. Ozonesondes are
typically launched once a week, although a few stations have more frequent launches. The aircraft
records generally contain more frequent observations, but observation times vary. For the selected
23 stations, we calculated the mean ozone vertical profiles at 1km resolution (the first layer is from
the surface to 1 km above sea level) for each month during the observational period for the two
datasets. A minimum of four aircraft profiles were required to estimate a monthly mean profile;
because ozonesonde launches are typically only a few times per month, no minimum was required
to estimate a monthly mean profile. Only data with monthly means in both datasets were included
for further analysis. Comparisons between the two datasets were made by ozonesonde type and by




station-airport distance.

**3. Results and discussion**
**3.1 Comparison of the vertical profiles of tropospheric $O_3$ from four types of ozonesondes and**
**aircraft observations**
Previous intercomparisons of sondes launched on the same balloon (Attmannspacher and Dütsch,
1970, 1981; Beekmann et al, 1994, 1995; Deshler et al., 2008; Hilsenrath et al., 1986; Kerr et al,
1994; Smit et al., 2007) have shown that sondes of different types respond somewhat differently to
the same ozone profile; that is, they have relative biases, that vary with altitude. Fig. 2 therefore
compares the mean vertical profiles of tropospheric $O_3$ from ozonesonde and aircraft measurements,
separated by ozonesonde type. Both $O_3$ concentrations and absolute differences between
ozonesonde and aircraft increase with altitude, especially above 9 km. Average tropospheric $O_3$
profiles observed by ECC, Brewer-Mast, and Carbon-Iodine sondes are in good agreement with
aircraft measurements, with biases of 7.03 ppb, 6.28 ppb and -4.48 ppb, respectively, and correlation
coefficients (R) of 0.72, 0.86 and 0.82, respectively (Fig. 3a-3c). The Indian-sonde average shows
a linear increase with altitude, while the aircraft measurements indicate an ozone decrease with
altitude above 8 km (Fig. 2b). This behaviour is most clearly related to the comparisons of stations
2°-4° apart in spring (Fig. S8). The agreement between the Indian-sonde and aircraft observations
is poor, with a bias of 24.07 ppb, and R of only 0.41 (Fig. 3d). The RMSE of $O_3$ observed with the
four types of ozonesondes (ECC, Brewer-Mast, Carbon-Iodine and Indian-sonde) and the aircraft is
small, at 49.15 ppb, 42.08 ppb, 33.55 ppb and 45.90 ppb, respectively.

Fig. 2 shows that the mean differences between ozonesonde and aircraft measurements vary

significantly with altitude. This can also be observed clearly from the relative differences (RD),
expressed as $(O_{3\text{-ozonesonde}} - O_{3\text{-aircraft}})/ O_{3\text{-aircraft}} \times 100\%$ (Fig. 4). $O_3$ concentrations from ECC
measurements are higher than those from aircraft measurements in all altitudes except at the surface.
Mean $O_3$ concentrations reported by Brewer-Mast sondes are lower than those from IAGOS below
7 km, but higher between 7 and 12 km. $O_3$ concentrations reported by Carbon-Iodine sondes are
higher than those observed from aircrafts below 2 km, but significantly lower above 8 km. In relative
terms, the bias between ECC sonde and aircraft measurements has only a very modest variation



with altitude, except near the ground. The mean relative bias for Brewer-Mast measurements is at
an absolute maximum of -19 % near the ground, but increases slowly above 3 km, and is positive
above 7 km, reaching more than +10 % at 10~11 km. The relative bias for Carbon-Iodine
measurements is about 8% below 2 km, is quite small from 2 – 8 km, and becomes large and negative
above ~8 km.
The Indian-sonde observations show much larger mean differences from the aircraft measurements.
Biases are everywhere positive, and as high as nearly 60% or 30 ppb, with much higher uncertainty
(standard errors) at each altitude as well (Fig. 2b, Fig. 4).
These results are broadly consistent with those from JOSIE 1996 (Smit et al. 1996; Smit and Kley,
1998; Thompson et al., 2019), and with the northern hemisphere average result from Tarasick et al.
(2019). (Their Figure 20b; note that it is largely based on ECC sondes, and the scale is inverted
(IAGOS-sondes) from the sense we use here.)
It should be noted that these comparisons only give an average relative bias between sondes and
IAGOS. The true value of the ozone profile remains unknown, as do the absolute biases of sondes
and IAGOS.

**3.2 Seasonal dependence of relative biases**
Fig. 5 compares mean profiles observed by ECC ozonesondes and IAGOS, separated by season.
There are modest seasonal differences in the relative bias profiles, with somewhat larger average
biases in winter and spring, but average biases are all positive (ECC sondes higher) and at all levels
the average seasonal biases are not statistically different at the 95% confidence level.
The modest seasonal differences that are apparent in Fig. 5 and in Figs. S1-S3 are likely due to the
modest sample size (for ECC sondes) and small sample sizes (for other types). The actual
coincidence in time for profiles can range from less than one day to about 1-3 weeks, depending on
the number of ozonesonde and aircraft $O_3$ profiles collected within each month-bin. This means the
larger the atmospheric variability of $O_3$ is, the larger the real differences between ozonesonde and
aircraft $O_3$ can become, particularly when the number of profiles within a month-bin are small. In
addition, there are errors due to variations in the aircraft take-off and landing trajectories and the
balloon rise rate, the geographical location of the observation stations (and any associated



meteorological differences) and any systematic difference in standard observational times.
Table 2 indicates that in all four seasons ECC data correlate well with aircraft observations, with R
ranging from 0.71 to 0.76, but with larger average biases in winter and spring, as noted. It is not
clear if these seasonal average differences in bias are significant, as the uncertainty ranges on the
seasonal averages (lower plot of Fig. 5) overlap.
The vertical distribution of tropospheric $O_3$ observed by Brewer-Mast and IAGOS aircraft in the
four seasons is similar (Fig. S1). Differences are also similar, except above 7 km, where the
uncertainties are larger, and in general the uncertainty ranges on the seasonal difference averages
overlap. Since these comparisons come from only one station pair, some of the differences may be
attributable to local differences in topography and meteorology. Table 2 shows that correlations for
the ensemble of Brewer-Mast stations are higher than those for ECC stations. Like the ECC sondes,
average biases are all positive, but this is determined by the biases above 7 km (Fig. 4); unlike the
ECCs, biases are negative in the lowest 3 km.
The vertical distribution of tropospheric $O_3$ concentrations observed by Carbon-Iodine sondes and
IAGOS aircraft in the four seasons are similar, except in summer when the tropopause is high (Fig.
S2). The difference plots are fairly similar, except in the lowest 3 km, where differences become
quite large in summer. Like the previous comparison for Brewer-Mast sondes, these comparisons
come from only one station pair, and so the large differences in the boundary layer in summer are
likely due to local ozone production sampled by the sonde but not the aircraft. Likely for this reason,
the consistency between Carbon-Iodine and aircraft observations is poor in summer, with R being
only 0.46 (Table 2). For the other three seasons it is fairly good.
The tropospheric $O_3$ observed by Indian-sondes displays a consistently high bias relative to IAGOS
in all seasons, and the seasonal difference plots are quite similar, except in the lowest 3 km in winter
(Fig. S3). This different behavior in winter is likely due to local ozone production sampled by the
aircraft but not the sonde. Temperature inversions are common in the winter in northern India and
trap local pollution. The very low values registered by the aircraft near the surface in summer also
suggest local effects, in this case titration by NOx.
Table 2 indicates poor consistency between Indian-sonde and aircraft observations in all four
seasons, with R in winter only 0.18.





### 3.3 Dependence of relative biases on station-airport distances

A major concern with comparing IAGOS and ozonesonde observations is that the stations and airports are not generally co-located, and even where they are close, the flight paths taken by balloon and aircraft are quite different. Fig. 6 compares the average vertical distribution of tropospheric $O_3$ observed at different station-airport distances by ECC sondes and IAGOS aircraft. Note that we continue to separate sonde station data by type --- only ECC data are used here. Sonde-aircraft pairs have been grouped by station-airport distance (Table 1). The differences in average bias vary only very modestly between the different station-airport distance categories, and those differences are not statistically different at the 95% confidence level (Fig. 6d). This, partially owing presumably to the use of mean monthly averages, is encouraging, as this provides further evidence that the average bias we have derived is an artifact strictly of instrument differences.

Table 3 indicates that the bias variation between ECC and aircraft observations at different station-airport distances is small, ranging from 5.7 ppb to 9.8 ppb. Correlations for these groupings are also fairly similar, at R = 0.8, 0.9 and 0.7.

Compared with ECC sondes, the consistency between the Indian-sonde and aircraft observations is poor at all station-airport distances, with much larger biases, and poor correlations, with R = 0.2 to 0.4. Nevertheless, Fig. S4 shows that the profiles of average differences are quite similar for station-airport distances < 1°, and distances of 2°~4° (Fig. S4c).

Fig. 7 and Figs. S5-S7 examine possible seasonal variation in the differences at different station-airport distances, for ECC sondes. The mean differences for the different station-airport distance categories are larger than for the annual averages (Fig. 6), but in general those differences are not statistically different at the 95% confidence level (Figs. 7d and S5d-S7d).

### 3.4 Comparison of ozonesonde relative biases under operational conditions using IAGOS observations as a transfer standard

The foregoing discussion demonstrates that, consistent with previous work, there is a fairly constant relative bias between IAGOS and sondes, with considerable dependence on sonde type, as expected from previous sonde intercomparisons like JOSIE 1996. Although uncertainties are sizeable due to the relatively sparse nature of the available data, we find consistent differences at all sites, with little



dependence on season or on station-airport separation, and little regional dependence (not shown).
Notwithstanding this overall sonde-IAGOS bias, we can use these station-airport comparisons to
derive relative biases of the different sonde types in use in the global network.
This does not assume that the aircraft data are unbiased. The true value of the ozone profile (or even
its average) remains unknown, as do the absolute biases of sondes and IAGOS. It does assume:
1. That the measurement errors are random and normally distributed;
2. That there is one, constant bias for each measurement type (that is, if, for example, the Indian
sonde has changed over the period of comparison, or the IAGOS instruments have different biases,
there would be additional error that is not included in our uncertainty estimate);
3. That the measurement biases are not dependent on the geographic location or other variability of
the ozone profile. This does not assume that the average ozone profile is the same, just that the
instruments respond in the same way.
With these assumptions we can use the results of Fig. 2 to estimate the relative biases of each
sonde type to each other. The uncertainty of the comparisons will be the quadratic sum of the
uncertainties of the two IAGOS-sonde comparisons. The results are shown in Table 4. This
intercomparison of the different sonde types has an important advantage: it compares ozonesonde
relative biases under operational conditions, as it compares the data that are actually in databases
like the WOUDC. It also fills a gap, as the last WMO international intercomparison involving all
four sonde types was JOSIE 1996. These results are broadly consistent with those from JOSIE 1996
(Smit and Kley, 1998; their Table 8 and Fig. 11).
In fact, the types of ozonesonde have changed during long-term observations at some stations (e.g.
Uccle and Payerne). De Backer et al. (1998) showed that with the use of an appropriate correction
procedure, accounting for the loss of pump efficiency with decreasing pressure and temperature, it
is possible to reduce the mean difference between ozone profiles obtained with both types of sondes
below 3%, which is statistically insignificant over nearly the whole operational altitude range (from
the ground to 32 km). Stübi et al. (2008) also found that the $O_3$ difference between the Brewer-Mast
and the ECC ozonesonde data shows good agreement between the two sonde types, and the profile
of the $O_3$ difference is limited to ±5% (±0.3 mPa) from the ground to 32 km. The results for Brewer-
Mast sondes in Table 4 should also be applicable to the older Payerne and Uccle records, and are



generally consistent with these results and with those for the older Canadian records (Tarasick et al.,

2002; 2016).

The results in Table 4 will be quite valuable for addressing the problem of relative biases when
merging ozonesonde data into global climatologies (e.g. McPeters et al., 2007; McPeters and Labow,
2012; Bodeker et al., 2013; Liu et al., 2013; Hassler et al., 2018;).
**4 Conclusions**
The vertical distribution of tropospheric $O_3$ observed by ozonesondes and IAGOS aircraft sensors
are compared at 23 pairs of sites between ~30°S and 55°N from 1995 to 2021. Overall, ECC,
Brewer-Mast, and Carbon-Iodine sondes agree reasonably well with aircraft observations, with
average biases of 7.03 ppb, 6.28 ppb, and -4.48 ppb, and correlation coefficients of 0.72, 0.86, and
0.82, respectively. The agreement between the aircraft and Indian-sonde observations is poor, with
an average bias of 24.07 ppb and R of 0.41.
Notwithstanding this general agreement, all sonde types show significant average biases with
respect to IAGOS. The $O_3$ concentration observed by ECC sondes is on average higher by 5-10%
than that observed by IAGOS aircraft, and the relative bias increases modestly with altitude.
Seasonal variations in the relative bias are not in general statistically significant at the 95%
confidence level. The distance between station and airport within 4° also has little effect on the
comparison results. When the ECC station pairs are grouped by station-airport distances of <1°
(latitude and longitude), 1-2°, and 2-4°, biases with respect to IAGOS measurements vary from 5.7
to 9.8 ppb, and correlations from 0.7 to 0.9.
Thus, the observed average relative bias between sondes and IAGOS found in this study, also noted
by previous authors (Zbinden et al., 2013; Staufer et al., 2013, 2014; Tanimoto et al., 2015; Tarasick
et al., 2019), is a robust result. Possible reasons for the difference include: side reactions that cause
sondes to produce excess iodine (Saltzman and Gilbert, 1959), and/or loss of ozone on the inlet
pump that could cause IAGOS monitors to read low at pressures below 800 hPa. The latter was an
issue in earlier aircraft ozone sampling programs (Schnadt Poberaj et al., 2007; Dias-Lalcaca et al.,
1998; Brunner et al., 2001), but Thouret et al. (1998) found it negligible for MOZAIC/IAGOS.
This result implies that care must be taken when merging ozonesonde and IAGOS measurement
datasets. While the aircraft and sonde measurements are often complementary, filling in important



spatial gaps that would otherwise exist if only one type were used, the records are not typically over
the same period, and so merging can introduce spurious jumps if relative biases are not taken into
account.
The importance of ozone in the troposphere as an air pollutant and a greenhouse gas, and therefore
of accurate measurements of its temporal and spatial distribution implies that it will be important to
resolve the causes of this bias, and so further research involving more direct comparisons of IAGOS
instrumentation and ozonesondes, e.g. in the WCCOS chamber, are strongly recommended.
These results are also useful to evaluate the relative biases of the different sonde types in the
troposphere, using the aircraft as a transfer standard. This intercomparison of the different sonde
types has the advantage that it compares ozonesonde relative biases under operational conditions;
that is, the data that are actually in databases like the WOUDC. These results will be invaluable for
addressing relative biases when merging ozonesonde data into global climatologies (e.g. Bodeker
et al., 2013; Hassler et al., 2018; Liu et al., 2013; McPeters et al., 2007; McPeters and Labow, 2012).

**Competing interests.** The contact authors have declared that none of the authors has any competing
interests.
**Data availability.** The global ozone sounding data were acquired from the World Ozone and
Ultraviolet Radiation Data Center (http://www.woudc.org) operated by Environment Canada. The
IAGOS data are created with support from the European Commission, national agencies in Germany
(BMBF), France (MESR), and the UK (NERC), and the IAGOS member institutions
(http://www.iagos.org/partners).
**Author contributions. HW**: Data curation, Methodology, Validation, Visualization, Writing -
original draft preparation, Writing - review & editing, Funding acquisition. **LS**: Methodology,
Investigation, Writing - original draft. **DWT**: Data curation, Resources, Conceptualization,
Supervision, Writing - original draft preparation, Writing - review & editing. **JL**: Data curation,
Resources, Methodology, Conceptualization, Supervision, Writing - original draft preparation,
Writing - review & editing, Funding acquisition. **TZ**: Funding acquisition, Writing - review &
editing. **HGJS** and **RVM**: Writing - review & editing.



**Acknowledgments.** We thank many whose dedication makes datasets used in this study possible.
The global ozone sounding data were acquired from the World Ozone and Ultraviolet Radiation
Data Center (http://www.woudc.org) operated by Environment Canada, Toronto, Canada, under the
auspices of the World Meteorological Organization. Flight-based atmospheric chemical
measurements are from IAGOS. IAGOS is funded by the European Union projects IAGOS-DS and
IAGOS-ERI. The IAGOS data are created with support from the European Commission, national
agencies in Germany (BMBF), France (MESR), and the UK (NERC), and the IAGOS member
institutions (http://www.iagos.org/partners). The participating airlines (Lufthansa, Air France,
Austrian, China Airlines, Iberia, Cathay Pacific, Air Namibia, Sabena) supported IAGOS by
carrying the measurement equipment free of charge since 1994. We are also thankful to the Digital
Research Alliance of Canada at the University of Toronto for facilitating data analysis.

**Financial support.** This study was supported by the Natural Science and Engineering Council of
Canada (Grant No. RGPIN-2020-05163), the National Key Research and Development Program of
China (Grant No., 2022YFC3701204), the National Natural Science Foundation of China
(42275196 and 41830965), the Natural Science Foundation of Jiangsu Province (BK20231300), and
Wuxi University Research Start-up Fund for Introduced Talents (2023r035).

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



# Figures

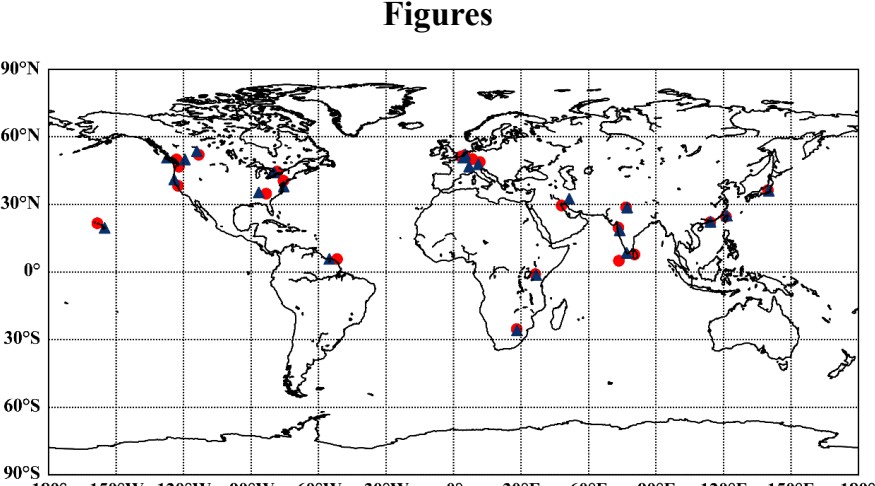

**Figure 1.** Map of 23 pairs of sites used in this study. Red circle markers are IAGOS sites, blue triangle markers are WOUDC sites.

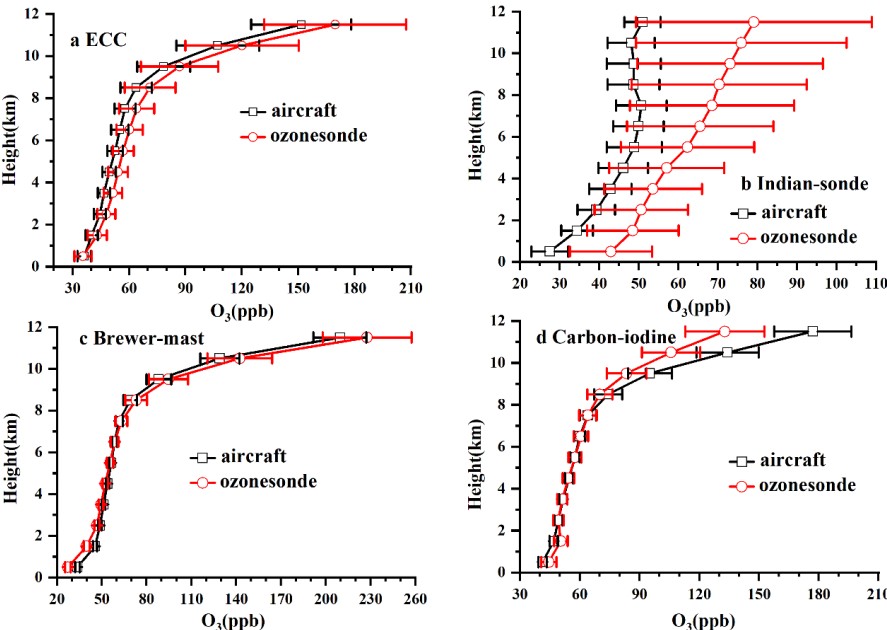

**Figure 2.** Comparison of the vertical profiles of tropospheric $O_3$ observed between aircraft measurements and four types of ozonesondes, ECC, Indian-sonde, Brewer-mast, and Carbon-iodine. The error bar length is 4 times the standard error (SE) of the mean (equivalent to 95% confidence limits on the averages).




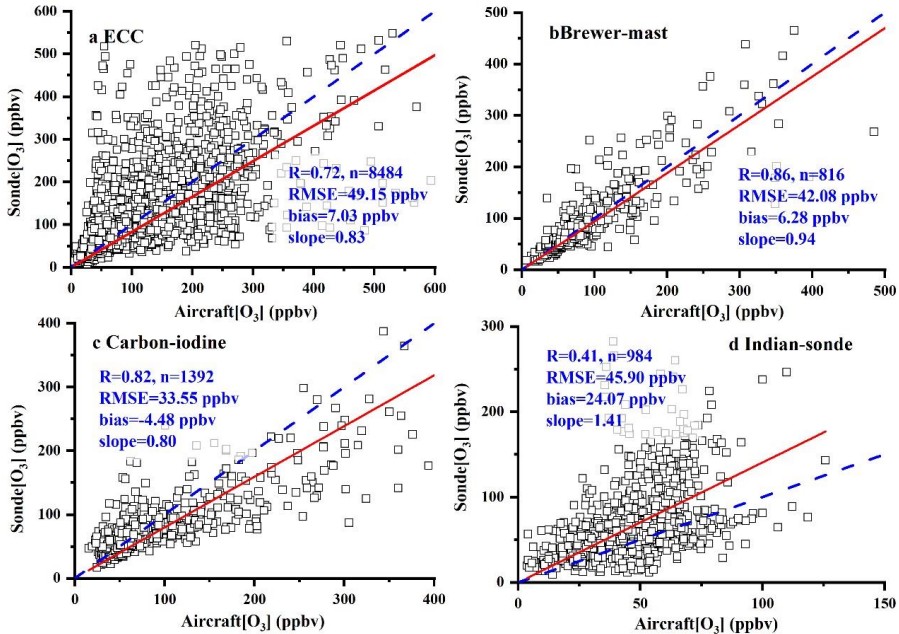

**Figure 3.** Correlation ($R$) of ozone mixing ratios between ozonesonde and aircraft measurements. The blue dashed line shows the 1:1 axis. Correlations are significant at the 99% level ($p < 0.01$). $N$ denotes the number of data points, and RMSE is the root mean square error.



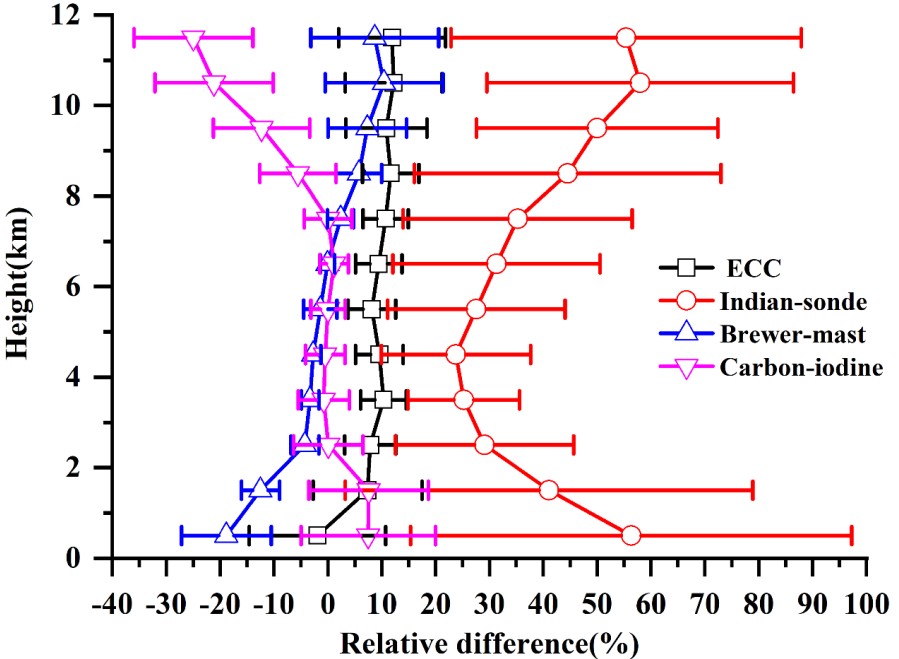

**Figure 4**. Mean relative difference (RD) between the ozonesonde $O_3$ and aircraft $O_3$ data. RD is calculated from $(O_{3\text{-ozonesonde}} - O_{3\text{-aircraft}})/ O_{3\text{-aircraft}} \times 100\%$.

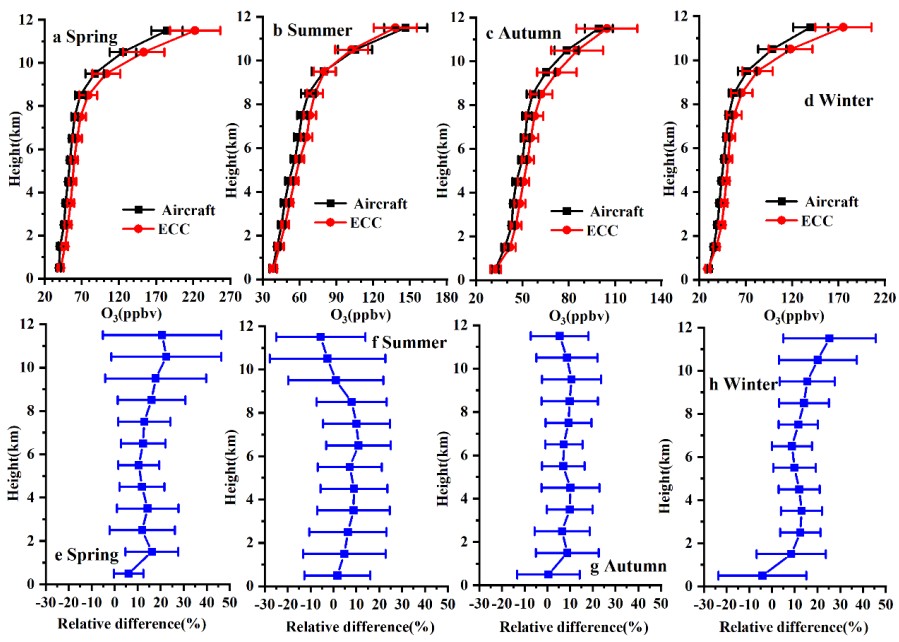



**Figure 5.** The mean difference in vertical profiles of the tropospheric $O_3$ between ECC ozonesonde and aircraft observations in four seasons (a-d) and their mean relative difference (e-h).

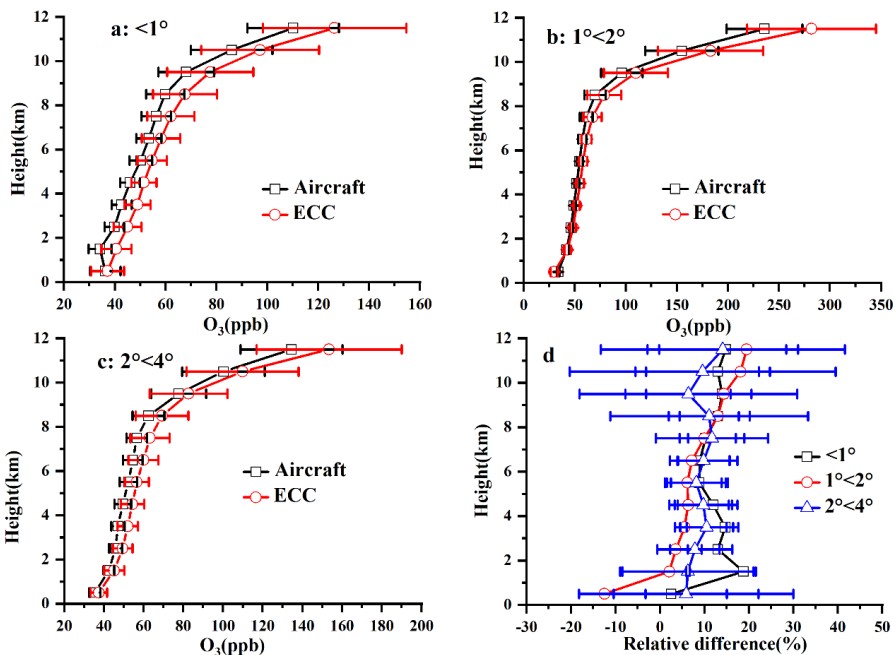

**Figure 6.** The annual mean vertical profiles of tropospheric $O_3$ between ECC ozonesonde and aircraft observations at station-pair distances (D) of D<1° (a), 1°< D <2° (b), and 2°< D <4°. The relative differences for the three categories are shown in (d).





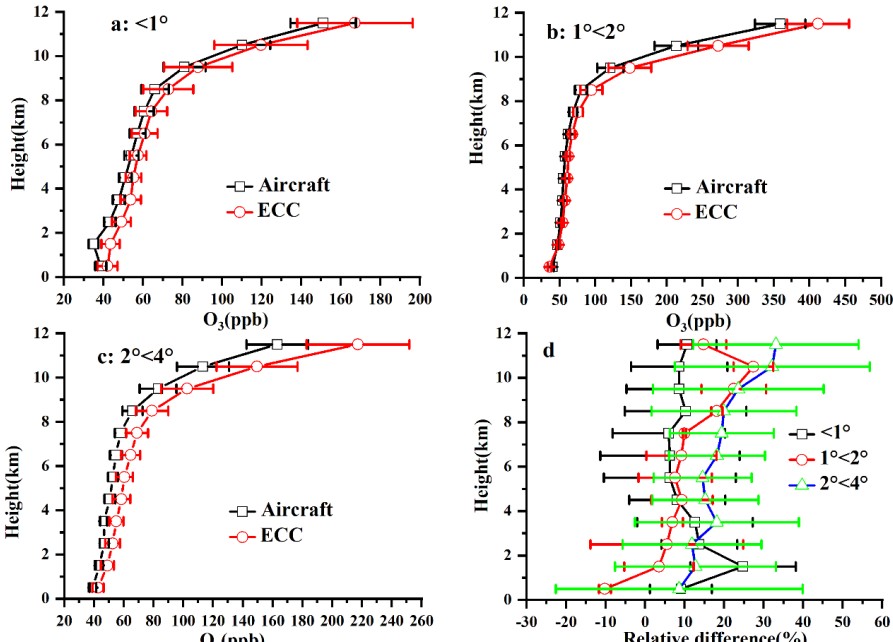

**Figure 7**. The seasonal mean vertical profiles of tropospheric $O_3$ in spring between ECC ozonesonde and aircraft observations at station-pair distances (D) of D<1° (a), 1°< D <2° (b), and 2°< D <4°.The relative differences for the three categories are shown in (d).



# Tables

**Table 1.** Summary of the station information, including station's name, geolocation, the number of profiles, observational period, and the station-pair distance used in this study.

| | MOZAIC-IAGOS | | | WOUDC | | | | | No. valid data months | observation period | station- airport distance |
|---|---|---|---|---|---|---|---|---|---|---|---|
| Station name | Lon | Lat | No. profiles | Station name | Lon | Lat | No. profiles | Type | | | |
| Toronto | -78.50 | 44.58 | 321 | Egbert | -79.78 | 44.23 | 181 | ECC | 33 | 2004-2008 | |
| Dusseldorf | 4.96 | 51.82 | 412 | De Bilt | 5.18 | 52.10 | 333 | ECC | 63 | 1995-2013 | |
| Munich | 11.63 | 48.84 | 2136 | Hohenpeissenberg | 11.01 | 47.80 | 1032 | Brewer-mast | 67 | 1996-2006 | |
| Johannesburg | 28.07 | -25.32 | 199 | Irene | 28.22 | -25.91 | 135 | ECC | 26 | 1998-2003 | |
| Nairobi | 36.33 | -0.94 | 114 | Nairobi | 36.75 | -1.30 | 42 | ECC | 10 | 1997-1998 | |
| Mumbai | 73.27 | 19.70 | 122 | Pune | 73.85 | 18.53 | 56 | Indian-sonde | 35 | 1996-2003 | <1° |
| Delhi | 76.65 | 28.73 | 342 | New Delhi | 77.18 | 28.63 | 88 | Indian-sonde | 50 | 1995-2016 | |
| Hongkong | 114.11 | 22.10 | 123 | King's Park | 114.17 | 22.31 | 115 | ECC | 25 | 2000-2005 | |
| Taipei | 121.08 | 24.59 | 2115 | Taipei | 121.48 | 25.02 | 58 | ECC | 31 | 2014-2018 | |
| Tokyo | 139.73 | 36.33 | 1342 | Tateno (Tsukuba) | 140.13 | 36.05 | 655 | Carbon-iodine | 116 | 1995-2006 | |
| Calgary | -113.25 | 52.03 | 170 | Edmonton | -114.10 | 53.55 | 112 | ECC | 17 | 2009-2011 | 1°~2° |
| Brussels | 3.24 | 51.21 | 2412 | Uccle | 4.36 | 50.80 | 736 | ECC | 55 | 1997-2009 | |
| Honolulu | -158.33 | 21.66 | 169 | Hilo (HI) | -155.07 | 19.58 | 107 | ECC | 16 | 2015-2017 | 2°~4° |




| | | | | | | | | | |
|---|---|---|---|---|---|---|---|---|---|
| Vancouver | -123.14 | 49.95 | 595 | Kelowna | -127.38 | 50.69 | 594 | ECC | 68 | 2003-2015 |
| San-Francisco | -122.50 | 38.30 | 34 | Trinidad Head (CA) | -124.15 | 41.05 | 53 | ECC | 10 | 1999-2001 |
| Portland | -122.06 | 46.76 | 385 | Kelowna | -119.38 | 49.97 | 317 | ECC | 45 | 2003-2009 |
| Atlanta | -83.28 | 34.78 | 34 | Huntsville (AL) | -86.58 | 35.28 | 85 | ECC | 10 | 1999-2006 |
| Washington | -75.59 | 40.52 | 610 | Wallops Island (VA) | -75.46 | 37.94 | 616 | ECC | 80 | 1994-2014 |
| Cayenne | -51.78 | 5.75 | 200 | Paramaribo | -55.21 | 5.81 | 64 | ECC | 9 | 2002-2013 |
| Frankfurt | 8.30 | 50.16 | 12742 | Payerne | 6.94 | 46.81 | 2673 | ECC | 204 | 2002-2020 |
| Kuwait-City | 48.01 | 29.52 | 105 | Esfahan | 51.43 | 32.48 | 34 | ECC | 17 | 2001-2004 |
| Male | 73.49 | 5.00 | 76 | Trivandrum | 76.95 | 8.48 | 45 | Indian-sonde | 24 | 1997-2000 |
| Colombo | 80.41 | 7.79 | 31 | Trivandrum | 76.95 | 8.48 | 37 | Indian-sonde | 11 | 1998-2000 |




**Table 2.** Bias, correlation coefficient (R), and RMSE for four types of ozonesonde and aircraft observations in four seasons.

| Type | Season | Bias (O$_{3\text{-ozonesonde}}$ - O$_{3\text{-aircraft}}$) (ppb) | R | RMSE (ppb) |
|---|---|---|---|---|
| ECC | Spring | 17.34 | 0.76 | 65.52 |
| | Summer | 1.96 | 0.76 | 40.15 |
| | Autumn | 1.75 | 0.71 | 34.47 |
| | Winter | 7.61 | 0.71 | 51.74 |
| Brewer-mast | Spring | 10.22 | 0.94 | 43.51 |
| | Summer | 2.99 | 0.83 | 48.79 |
| | Autumn | 6.53 | 0.79 | 29.40 |
| | Winter | 6.11 | 0.88 | 45.45 |
| Carbon-iodine | Spring | -9.19 | 0.84 | 38.34 |
| | Summer | 3.83 | 0.46 | 29.31 |
| | Autumn | 2.33 | 0.68 | 15.10 |
| | Winter | -16.68 | 0.88 | 44.72 |
| Indian-sonde | Spring | 19.64 | 0.44 | 44.30 |
| | Summer | 19.58 | 0.57 | 37.44 |
| | Autumn | 20.38 | 0.45 | 37.30 |
| | Winter | 40.07 | 0.18 | 64.99 |

**Table 3**. Bias, correlation coefficient(R) and RMSE for ECC and Indian-sonde ozonesonde and aircraft observations at different station-airport distances.

| Type | Station-pair distance | Bias (O$_{3\text{-ozonesonde}}$ - O$_{3\text{-aircraft}}$) (ppb) | R | RMSE (ppb) |
|---|---|---|---|---|
| ECC | <1° | 9.78 | 0.78 | 47.46 |
| | 1°~2° | 8.91 | 0.90 | 40.73 |
| | 2°~4° | 5.65 | 0.67 | 51.00 |
| Indian-sonde | <1° | 26.71 | 0.37 | 49.54 |
| | 2°~4° | 15.35 | 0.24 | 30.86 |





**Table 4**. Comparison of the sondes of each type to IAGOS. (average ± 2 times the standard error (SE)) Indian-sonde/ECC is (Indian-sonde/IAGOS)/(ECC/IAGOS), Brewer-mast/ECC is (Brewer-mast/IAGOS)/(ECC/IAGOS), Carbon-iodine/ECC is (Carbon-iodine /IAGOS)/(ECC/IAGOS)

| Altitude(km) | Indian-sonde/ECC | Brewer-mast/ECC | Carbon-iodine/ECC | ECC/ IAGOS |
|---|---|---|---|---|
| 0~1 | 1.59 ±1.74 | 0.83 ±0.96 | 1.10 ±1.36 | 0.98 ±1.28 |
| 1~2 | 1.31 ±1.83 | 0.81 ±0.90 | 1.00 ±1.05 | 1.07 ±1.58 |
| 2~3 | 1.20 ±1.62 | 0.89 ±0.97 | 0.93 ±0.85 | 1.08 ±1.54 |
| 3~4 | 1.14 ±1.57 | 0.88 ±0.94 | 0.90 ±0.87 | 1.10 ±1.48 |
| 4~5 | 1.13 ±1.61 | 0.89 ±1.02 | 0.91 ±0.99 | 1.10 ±1.44 |
| 5~6 | 1.18 ±1.76 | 0.91 ±1.05 | 0.92 ±1.04 | 1.08 ±1.37 |
| 6~7 | 1.20 ±1.89 | 0.91 ±1.00 | 0.92 ±0.82 | 1.09 ±1.54 |
| 7~8 | 1.22 ±1.92 | 0.92 ±0.94 | 0.90 ±0.64 | 1.11 ±1.69 |
| 8~9 | 1.29 ±2.09 | 0.95 ±0.99 | 0.85 ±0.55 | 1.12 ±1.61 |
| 9~10 | 1.35 ±2.35 | 0.97 ±1.09 | 0.79 ±0.62 | 1.11 ±1.46 |
| 10~11 | 1.41 ±3.26 | 0.98 ±1.21 | 0.70 ±0.68 | 1.12 ±1.37 |
| 11~12 | 1.39 ±4.61 | 0.97 ±1.19 | 0.67 ±0.72 | 1.12 ±1.42 |

