# Peer review of "Consistency evaluation of tropospheric ozone from ozonesonde and"

_EGUsphere, 2024_

## Author Comment (AC1)

**Review 1**

The paper gives a comprehensive update on the comparison of long-term ozone data from balloon-borne ozone sondes and IAGOS measurements on commercial aircraft. As in previous studies, the authors find that ECC ozone sondes give 5 to 10% higher ozone values in the free troposphere (3 to 8 km). Brewer-Mast and Carbon-Iodine sondes give slightly lower ozone, 0 to 5% less, in the free troposphere. Monthly means from all three sonde types show high correlations, more than 0.7, with IAGOS monthly means. Indian ozone sondes usually give 25 to 35% higher ozone than IAGOS, with generally poor correlation, around 0.4. The authors find little to no dependence on season or distance between sonde station and IAGOS airport. Overall this is a well written paper which is relevant and should be published in ACP.

Response: We appreciate the time and effort that the editor and the reviewers dedicated to providing feedback on our manuscript. We are grateful for the insightful and valuable comments on our paper. We have incorporated most of the suggestions made by the reviewers. Those changes are highlighted in the revised manuscript. According to your suggestion, we modified the manuscript in detail and marked the revised contents with red font.

I do have a number of generally minor suggestions:

For several parts of the paper, I would prefer a clearer separation between three altitude regions and would like to see more specific results for these altitude regions. In many places, e.g. Fig. 3, there is much better agreement for the 3 to 8 km region, less agreement for altitudes below 3 km and above 8 km. The region below 3 km has a lot of local ozone sources and sinks (cities, airports, rural environment, ...), while the region above 8 km is influenced quite significantly be stratosphere-tropopsphere exchange, jet-streams, tropopause folds, .. Separating results for these regions would provide a clearer picture of ozone-sonde and IAGOS differences, as well as the limitations of the current comparison.

Response: Thanks for your suggestions. We added: " **Figure S1**. Bias, correlation coefficient (R), and RMSE for four types of ozonesonde and aircraft observations in different altitudes."

[Figure]

**Figure S1**. Bias, correlation coefficient (R), and RMSE for four types of ozonesonde and aircraft observations in different altitudes.

And added in the lines 201-224:

The region below 3 km has many local ozone sources and sinks (cities, airports, rural environment, etc). In comparison, the region above 8 km is significantly influenced by stratosphere-troposphere exchange, jet streams, and tropopause folds. Fig. S1 shows that the R between ozonesondes and aircraft observations is higher near the ground (< 2 km) and at high altitudes (> 10 km). This shows that although the influencing factors of $O_3$ near the ground and at high altitudes are more complex, their long-term temporal variation characteristics are similar. The influences of cities, airports, rural environment, stratosphere-troposphere exchange, jet streams, tropopause folds, etc., have a more significant impact on the concentration of $O_3$ in the short term.

The correlation between four types of ozonesondes and aircraft observations also varies with altitude (Fig. S1). From 0-8 km, the correlation between ECC and aircraft observations decreases with altitude, with R being 0.71 at 0-1 km and reaching a minimum of 0.29 at 8-9 km; from 8-12 km, R increases with altitude, reaching 0.49 at 11-12 km. The correlation between the other three

ozonesondes (Brewer-mast, Indian-sonde and Carbon-iodine) and the aircraft observations all vary with altitude, with different inflection points. The number of stations for these three types of ozonesondes is small (Table 1). Therefore, local variable influences on $O_3$ are more important, so R varies more with altitude.

The bias and RMSE with respect to the aircraft observations of the four types of ozonesondes at 8-12 km are higher than that at other altitudes. In contrast, the bias and RMSE values below 8 km are smaller and vary less with altitude, consistent with the vertical distribution characteristics of $O_3$ concentration in Fig. 2. This is likely due to the higher concentration of $O_3$ and the typically larger difference in spatial distance between ozonesonde and aircraft observations at 8-12km.

In addition, the bias and RMSE relative to the aircraft observations at different altitudes for ECC, Carbon-iodine and Brewer-mast sondes are lower than those for the Indian-sonde, which is similar to the results of the above analysis of $O_3$ concentration.

And in conclusions, we added "Ozonesondes and aircraft observations have smaller R in the middle troposphere, but larger bias and RMSE in the upper troposphere. The bias and RMSE relative to the aircraft observations at different altitudes for ECC, Carbon-iodine and Brewer-mast sondes are lower than those for the Indian-sonde."

Table 1: station - airport distance is missing in most cases. Should be given, preferably also in kilometers. Also: minus sign before longitudes in the western hemisphere ended up in a different line. Please fix.

Response: Thanks for your suggestions. We have modified it and added the station-airport distance (km) in Table 1.

Figure 3: please state in the caption that this is based on monthly means for both sondes and IAGOS. Also: It would be better to see the relative frequency of the data points, e.g. using a false color representation. As it is now, the plot tends to emphasize the more outlying data points, and one cannot see where most of the datapoints lie. I assume close to the fitted red lines. Also: I find it confusing that the fitted lines are in red, while the text describing the fits is in blue and in the same color as the 1:1 line. Please make these colors consistent. Finally: In Fig. 2d, the Indian sonde data are clearly higher than the IAGOS data. In contrast, in Fig. 3d, the fitted line is below the 1:1 line, which would indicate that the sondes give lower ozone, in contradiction to Fig. 2d. I think it would

be better to not force the fitted lines through zero, but fit slope and offset. An offset could indicate potential causes for systematic differences, e.g., high background current in the sonde data.

Response: Thanks for your suggestions. We have redrawn Figure 3. We found that the force the intercept to zero for the regressions and not force the intercept to zero for the regressions are quite different, just as the reviewer said, "not force the fitted lines through zero, but fit slope and offset. An offset could indicate potential causes for systematic differences, e.g., high background current in the sonde data." But in generally, when $O_3$ is zero both the ozonesondes and the aircraft will measure zero. Therefore, we only give the force the intercept to zero for the regressions in the article, but we give the slope and offset of the not force the intercept to zero for the regressions in the main text analysis.

[Figure]

We added in Line 178-185: " After calculation, we obtained the slopes and offsets of ECC, Brewer-mast, Carbon-iodine and Indian-sonde without forcing the fitted lines through zero, the slope is 0.71, 0.88, 0.56 and 0.74, respectively, and the offset is 18.94 ppb, 6.89ppb, 27.48ppb and 27.84ppb. When we force the intercept to zero for the regressions, the slope is larger than the slope without forcing the fitted lines through zero (fig. 3). In generally, when $O_3$ is zero both the ozonesondes and the aircraft will measure zero. However, there is an offset in the fit of the two data sets due to

potential causes for systematic differences during the observation measurement process, e.g., high background current in the sonde data. "

[Figure]

**Figure 3.** Correlation (*R*) of monthly mean ozone mixing ratios between ozonesonde and aircraft measurements. While IAGOS does measure in the lower stratosphere these values are usually far from the airport, so the sonde-aircraft distance will be large, we only plots data below 150 ppb. The black dashed line shows the 1:1 axis, the red line shows the linear fit (with the intercept set to 0), the color bar shows the data counts. Correlations are significant at the 99% level (*p* <0.01). *N* denotes the number of data points, R is the correlation coefficient, Bias is the overall average difference in monthly mean values [Ozonesonde ozone – Aircraft ozone, in ppb], RMSE is the root mean square error, slope is the slope of the linear fit line. All data points are based on the monthly mean.

Line 166: I would not call a ~45ppb RMSE "small". Tropospheric ozone mixing ratios are around 50 ppb, so 45 ppb RMSE corresponds to around 100% uncertainty. That is hardly "small". Please correct.

Response: Thanks for your suggestions. We modified into " The RMSE of $O_3$ observed with the four types of ozonesondes (ECC, Brewer-Mast, Carbon-Iodine and Indian-sonde) and the aircraft is 15.99 ppb, 14.15 ppb, 16.26 ppb and 29.85 ppb, respectively."

Figure 4: Please add zero line for easier reference. Also, Fig 3 and Fig 4 bring up the question how R, slope, offset and RMSE behave for different altitudes. I think this should be considered, and additional plots should be shown and discussed. I have a feeling that agreement would be best near 5 km, and would deteriorate significantly below 3 km and above 9 km.

Response: Thanks for your suggestions. We have added the zero line in Figure 4. We have added "Figure S1 Bias, correlation coefficient (R), and RMSE for four types of ozonesonde and aircraft observations in different altitudes" and performed the analysis in the article. Please see the response to comment 1 for details.

[Figure]

**Figure 4**. Mean relative difference (RD) between the ozonesonde $O_3$ and aircraft $O_3$ data. RD is calculated from ($O_3$-ozonesonde - $O_{3\text{-aircraft}}$)/ $O_{3\text{-aircraft}} \times 100\%$. The green dashed line is the zero line.

We also plotted the fitting figures for 0-3 km, 3-8 km and 8-11 km, but we found that the R value at 0-3 km is relatively high. This indicates that although near-ground $O_3$ is greatly affected by the underlying surface and anthropogenic sources, this influence exists for both ozonesondes and aircraft, that is, the influence on these two observation datasets is consistent, so when comparing their temporal correlation, the influence is not significant.

[Figure]

Figure 1 Correlation (*R*) of monthly mean ozone mixing ratios between ozonesonde and aircraft measurements at 0-3 km. The blue dashed line shows the 1:1 axis and the red line shows the linear fit. Correlations are significant at the 99% level (*p* <0.01). *N* denotes the number of data points, and RMSE is the root mean square error.

[Figure]

Figure 2 Correlation (*R*) of monthly mean ozone mixing ratios between ozonesonde and aircraft measurements at 3-8 km. The blue dashed line shows the 1:1 axis and the red line shows the linear fit. Correlations are significant at the 99% level (*p* <0.01). *N* denotes the number of data points, and RMSE is the root mean square error.

[Figure]

Figure 3 Correlation (*R*) of monthly mean ozone mixing ratios between ozonesonde and aircraft measurements at 8-11 km. The blue dashed line shows the 1:1 axis and the red line shows the linear fit. Correlations are significant at the 99% level (*p* <0.01). *N* denotes the number of data points, and RMSE is the root mean square error.

Along the same lines, a table similar to Table 2, but giving results not for the four season but for three (or more) altitude regions would be very helpful.

Response: Thanks for your suggestions. We have added "Figure S1 Bias, correlation coefficient (R), and RMSE for four types of ozonesonde and aircraft observations in different altitudes" and performed the analysis in the article. Please see the response to comment 1 for details.

Table 4: Please clarify what is shown: Ratio of three other sonde types to ECC sondes, using IAGOS as transfer standard. Instead of showing X/ECC, it might be better to show X/ECC-1 and difference values as percent. Also: are the given uncertainties / standard errors correct. For Brewer-Mast/ECC at 0~1km, the given ratio is 0.83+-0.96. That means the ratio could be anywhere between -0.13 and 1.79. Really that wide uncertainty range? Please check.

Response: Thanks for your suggestions. Table 4 is the "Ratio of three other sonde types to ECC sondes, using IAGOS as a transfer standard." We intend to give such ratios so that we can use these ratios to convert different types of ozonesonde data. In addition, the values in Table 4 are average ± 2 times the standard error (SE), that is, the 95% confidence limts. Unfortunately, they are that large. not the standard deviation.

Discussion at the end, around lines 310 to 320: Could the high ozone observed by sondes have something to do with insufficient background subtraction? Certainly might be a problem for the Indian sondes who are also flying in a region with low tropospheric ozone. What would be the implication of the new improved background estimation methods outlined by Vömel et al. 2020 and Smit et al. 2023? Please discuss.

Response: Thanks for your suggestions. Yes, and no; background subytaction is not much of an issue at northern midlatitudes (where most of our comparison data originate). We added instead" However, as noted by Saltzman and Gilbert (1959), the differences in stoichiometry found at different pH values imply that the chemistry of reaction of ozone with KI is complex, involving reactions that cause loss of iodine, as well as reactions other than the principal one that produce additional iodine. Several authors have noted the existence of slow side reactions involving the phosphate buffer, with a time constant of about 20 minutes, that may also increase the stoichiometry from 1.0 (Tarasick et al., 2021, Smit et al., 2024). Furthermore, evaporation causes the concentration of the sensing solution to increase, which can further enhance the stoichiometry, by concentrating the phosphate buffer, and to a lesser degree, by increasing the concentration of the KI itself (Johnson et al., 2002). These factors could contribute to the observed average relative bias between sondes and IAGOS found in this study."

Since we do not know the preparation and sampling process or data reduction scheme of the Indian-sonde, and there are no recent intercomparison results, we cannot speculate on the impact of stoichiometry or background subtraction on the Indian-sonde results.

Also: Is there no new information since Thouret et al. 1998 checking for the correctness of the MOZAIC / IAGOS inlet system? Is it possible that enhanced $NO_x$ from aircraft exhaust gases results in local reductions (titration) of ozone in aircraft flight corridors? Local differences in the photo-chemical ozone regime ($NO_x$ limited or VOC limited) could very well play a role for differences between sonde stations and airports in the lowest 1 to 3 kilometers of the atmosphere.

Response: Thanks for your suggestions. We have now included unpublished results of a recent intercomparison of IAGOS instruments at the World Calibration Centre for Ozone Sondes (WCCOS) in Julich in June 2023. According to the publicly available literature, no one has systematically compared the consistency of MOZAIC/IAGOS observations since Thouret et al. 1998. Enhanced NOx from aircraft exhaust gases may result in local ozone reductions (titration) in aircraft flight

corridors. Unfortunately, we do not have relevant NOx data. Although IAGOS has NOx observation data, the observation data is relatively short-term, and the amount of data is small. When we compared the two data sets in our article, we first processed the two observation data sets into monthly average data. Therefore, what we compared was the consistency of the monthly average data of the two data sets. For the monthly average $O_3$ data, the impact of NO titration and local differences in the photo-chemical ozone regime (NOx limited or VOC limited) is relatively small, which is not the main reason for the difference between the two data sets.

**References:**

Smit, H. G. J., Poyraz, D., Van Malderen, R., Thompson, A. M., Tarasick, D. W., Stauffer, R. M., Johnson, B. J., and Kollonige, D. E.: New insights from the Jülich Ozone Sonde Intercomparison Experiment: calibration functions traceable to one ozone reference instrument, Atmos. Meas. Tech., 17, 73–112, https://doi.org/10.5194/amt-17-73-2024, 2024.

Vömel, H., Smit, H. G. J., Tarasick, D., Johnson, B., Oltmans, S. J., Selkirk, H., Thompson, A. M., Stauffer, R. M., Witte, J. C., Davies, J., van Malderen, R., Morris, G. A., Nakano, T., and Stübi, R.: A new method to correct the electrochemical concentration cell (ECC) ozonesonde time response and its implications for "background current" and pump efficiency, Atmos. Meas. Tech., 13, 5667–5680, https://doi.org/10.5194/amt-13-5667-2020, 2020.

---

## Author Comment (AC2)

Consistency evaluation of tropospheric ozone from ozonesonde and IAGOS aircraft observations: vertical distribution, ozonesonde types and station-airport distance. IAGOS and WOUDC are the two global $O_3$ vertical observation programs with the most observation sites and the longest continuous observation time in the world. The $O_3$ vertical observation data obtained are widely used in verifying models and satellites and analyzing $O_3$ vertical distribution characteristics, but systematic comparisons of these two data sets are still very rare. This paper selected 23 pairs of sites between about 30°S and 55°N from 1995 to 2021 for a detailed comparative analysis and discusses the vertical distribution and the influence of ozonesonde types and station-airport distance of the two data sets. The authors obtained relatively good comparison results: "The $O_3$ concentration observed by ECC sondes is on average higher by 5-10% than that observed by IAGOS aircraft, and the relative bias increases modestly with altitude." and "The distance between station and airport within 4° has little effect on the comparison results." The paper's topic is well suited for Atmospheric Chemistry and Physics, and the results are interesting. However, some important information is missing, and several issues need to be revised. Therefore, a minor revision is necessary before it can be accepted.

Response: We appreciate the time and effort that the editor and the reviewers dedicated to providing feedback on our manuscript. We are grateful for the insightful comments and valuable improvements to our paper. We have incorporated most of the suggestions made by the reviewers. Those changes are highlighted in the revised manuscript. According to your suggestion, we modified the manuscript in detail and marked the revised contents with red font.

1. abstract: There is not much content in the abstract, so it does not need to be divided into two paragraphs.

Response: Thanks for your suggestions. We have modified it.

2. line 83-87, In the introduction, it is recommended that the authors add some content related to the research significance of this study. Although relevant elaboration has been made in the literature review part of the introduction, it is still necessary to create a more explicit discussion on the research significance of this article in the last paragraph of the introduction.

Response: Thanks for your suggestions. We modified it into "As shown above, the global $O_3$ vertical distribution datasets observed by WOUDC and IAGOS have been widely used in various studies.

Still, a long-term and multi-site systematic comparison of these two datasets is rare, especially for the observations in the past three decades. In this study, we attempt to make the most comprehensive evaluation to date of the relative biases of IAGOS and sonde profiles, using as many station pairs as possible. We identify 23 suitable pairs of sites in the WOUDC and IAGOS datasets from 1995 to 2021, compare the average vertical distribution of tropospheric $O_3$ shown by ozonesonde and aircraft measurements, and analyze their differences by ozonesonde type and by station-airport distance."

3. In lines 109-110, only the $O_3$-related content needs to be introduced.

Response: Thanks for your suggestions. We deleted the content unrelated to $O_3$ and modified it to: "Blot et al. (2021) evaluated the internal consistency of the $O_3$ measurements since 1994, which confirmed the instrumental uncertainty of $\pm2$ ppb. Moreover, they found no bias drift amongst the different instrument units (six $O_3$ IAGOS-MOZAIC instruments, nine IAGOS-Core Package1 and the two instruments used in the IAGOS-CARIBIC aircraft)."

4. In Table 1, the layout is not very nice, as the minus sign before the western hemisphere longitude appears in different rows.

Response: Thanks for your suggestions. We have modified it.

5. There are some formatting problems in the article, which need to be carefully checked and modified, such as the mixed-use of "-" and "~".

Response: Thanks for your suggestions. We have modified them.

6. Line 231-232: Can you provide a more detailed discussion on the analysis of Indian-sonde? This part seems to be just a simple summary of the content in one sentence. Although the comparison between Indian-sonde and Aircraft is not good, it can be found from Table 2 that there are still some seasonal differences.

Response: Thanks for your suggestions. In lines 269-279, we modified into " The tropospheric $O_3$ observed by Indian-sonde in the four seasons is 43.3-79.4 ppb, 31.4-80.2 ppb, 42.2-69.6 ppb and 51.5-87.5 ppb, and that observed by aircraft in the four seasons is 22.8-60.1 ppb, 14.8-47.1 ppb, 25.0-44.1 ppb and 35.6-53.3 ppb (Fig. S4). The tropospheric $O_3$ observed in Indian-sonde in the four seasons increases with height almost linearly. The tropospheric $O_3$ observed by aircraft first increases and then decreases with altitude in spring, summer and autumn, while in winter, it first decreases and then increases with altitude. The tropospheric $O_3$ observed by the Indian-sonde and

the aircraft is quite different, and the RD in the four seasons is 6.3% to 47.5%, 22.6% to 52.9%, 26.4% to 40.6% and 5.13% to 39.13%. Table 2 indicates poor consistency between Indian-sonde and aircraft observations in all four seasons, with R in winter only 0.18. The bias and RMSE in winter are the largest, at 40.07 ppb and 64.99 ppb. The bias, R and RMSE in the other three seasons are smaller, and the differences between them slight."

---

## Author Comment (AC3)

Comments from the IAGOS PIs in charge of the IAGOS Ozone data set, its measurement on board passenger aircraft and its long-term quality.

It is not our intention to substitute ourselves to the work of the nominated reviewers regarding the overall scientific quality of the study. However, one major flaw is the comparison of the ozonesondes within 4 degrees of the IAGOS airports. It is difficult to make any conclusions from this due to the ozonesondes and IAGOS aircraft sampling completely different air masses. Because the authors did not respect the IAGOS data policy (https://iagos.aeris-data.fr/data-policy/), we did not have the opportunity to discuss this manuscript before submission. It was mandatory to inform the IAGOS Ozone PIs (from CNRS).

Therefore, the main purpose of this comment is to warn co-authors, reviewers, as well as the editors, and potential readers that this manuscript does not address potential defaults of the IAGOS ozone data sets. (i) none of the co-authors are committed to delivering the high quality of the IAGOS ozone data sets, and (ii) lines 314-315 are wrong. An intercomparison campaign in Julich in June 2023 demonstrated that there is no influence of the pumps on the Ozone IAGOS measurements between 1000 and 200 hPa. This is mentioned in Line 326, as a future activity. However, this activity was completed last year and the co-authors know this. The results, under preparation for publication, show that the IAGOS-CORE ozone measurements (Package 1 with pressurization pumps) and IAGOS-CARIBIC ozone measurements differ by less than 2% and the WCCOS reference UV photometer measurements are usually higher by maximum 5% compared to both IAGOS instruments.

Response: We appreciate the constructive comments by Dr. Valérie Thouret. In our manuscript, we have provided the following acknowledgments section in accordance with IAGOS requirements: "**Acknowledgments.** We thank many whose dedication makes datasets used in this study possible. The global ozone sounding data were acquired from the World Ozone and Ultraviolet Radiation Data Center (http://www.woudc.org) operated by Environment Canada, Toronto, Canada, under the auspices of the World Meteorological Organization. Flight-based atmospheric chemical measurements are from IAGOS. IAGOS is funded by the European Union projects IAGOS-DS and IAGOS-ERI. MOZAIC/CARIBIC/IAGOS data were created with support from the European Commission, national agencies in Germany (BMBF), France (MESR), and the UK (NERC), and

the IAGOS member institutions (http://www.iagos.org/partners). The participating airlines (Lufthansa, Air France, Austrian, China Airlines, Hawaiian Airlines, Air Canada, Iberia, Eurowings Discover, Cathay Pacific, Air Namibia, Sabena) supported IAGOS by carrying the measurement equipment free of charge since 1994. The data are available at http://www.iagos.fr thanks to additional support from AERIS. We are also thankful to the Digital Research Alliance of Canada at the University of Toronto for facilitating data analysis."

We are very sorry that we failed to consult the IAGOS PIs before submitting the manuscript. Therefore, we contacted the IAGOS PIs immediately after the reviewer pointed out this problem. Once again, we sincerely apologize to the IAGOS PIs for our negligence and hope the reviewer can forgive our unintentional mistake.

The reason for comparing the ozonesondes within 4 degrees of the IAGOS airports is to make a more systematic comparison between the two datasets. However, we only screened out ten sites with a distance < 1°, which we believe is insufficient to reflect the two datasets' global observations. Therefore, we expanded the distance range of the site selection to 23 sites, enhancing our comparison results' credibility. In addition, it is worth noting that our comparison found: "For the ECC ozonesonde, the overall bias with respect to IAGOS measurements varies from 5.7 to 9.8 ppb, when the station pairs are grouped by station-airport distances of <1° (latitude and longitude), 1-2°, and 2-4°. Correlations for these groups are R = 0.8, 0.9 and 0.7." This indicates that differences in the distance of the observation sites does not have a large impact on the comparison results, which is encouraging, as it is what we would expect for a constant instrument-related bias.

Lines 314-315 are wrong. It is important to note that the "and/or loss of ozone on the inlet pump that could cause IAGOS monitors to read low at pressures below 800 hPa" mentioned in our article refers to a problem that only existed in early aircraft observations, which is clearly stated in our article. "Possible reasons for the difference include: side reactions that cause sondes to produce excess iodine (Saltzman and Gilbert, 1959), and/or loss of ozone on the inlet pump that could cause IAGOS monitors to read low at pressures below 800 hPa. The latter was **an issue in earlier aircraft ozone sampling programs** (Schnadt Poberaj et al., 2007; Dias-Lalcaca et al., 1998; Brunner et al., 2001), **but Thouret et al. (1998) found it negligible for MOZAIC/IAGOS."**

In Lines 367-385, we added, " A recent intercomparison campaign at the World Calibration Centre

for Ozone Sondes (WCCOS) in Julich in June 2023 indicates that the pumps do not greatly influence the ozone IAGOS measurements between 1000 and 200 hPa. The IAGOS-CORE ozone measurements (Package 1 with pressurization pumps) and IAGOS-CARIBIC ozone measurements differ by less than 2%, and the WCCOS reference UV photometer measurements are usually higher by 1-2% (to a maximum of 5%) compared to both IAGOS instruments (Blot et al., 2021; Nédélec et al., 2015; Thouret et al., 2022). IAGOS-CARIBIC does not have pressurization system, so that's why the good comparison between both IAGOS systems means a lot.

However, as noted by Saltzman and Gilbert (1959), the differences in stoichiometry found at different pH values imply that the chemistry of reaction of ozone with KI is complex, involving reactions that cause loss of iodine, as well as reactions other than the principal one that produce additional iodine. Several authors have noted the existence of slow side reactions involving the phosphate buffer, with a time constant of about 20 minutes, that may also increase the stoichiometry from 1.0 (Tarasick et al., 2021, Smit et al., 2024). Furthermore, evaporation causes the concentration of the sensing solution to increase, which can further enhance the stoichiometry, by concentrating the phosphate buffer, and to a lesser degree, by increasing the concentration of the KI itself (Johnson et al., 2002). These factors could contribute to the observed average relative bias between sondes and IAGOS found in this study. "

Added references:

Blot, R., Nedelec, P., Boulanger, D., Wolff, P., Sauvage, B., Cousin, J.-M., Athier, G., Zahn, A., Obersteiner, F., Scharffe, D., Petetin, H., Bennouna, Y., Clark, H., Thouret, V., 2021. Internal consistency of the IAGOS ozone and carbon monoxide measurements for the last 25 years. Atmospheric Measurement Techniques, 14, 3935-3951, https://doi.org/10.5194/amt-14-3935-2021.

Nédélec, P., Blot, R., Boulanger, D., Athier, G., Cousin, J.M., Gautron, B., Petzold, A., Volz-Thomas, A., Thouret, V., 2015. Instrumentation on commercial aircraft for monitoring the atmospheric composition on a global scale: the IAGOS system, technical overview of ozone and carbon monoxide measurements. Tellus B: Chemical and Physical Meteorology, 67(1), 27791. https://doi.org/10.3402/tellusb.v67.27791@zelb20.2016.68.issue-s1

Smit, H. G. J., Poyraz, D., Van Malderen, R., Thompson, A. M., Tarasick, D. W., Stauffer, R. M., Johnson, B. J., and Kollonige, D. E., 2024. New insights from the Jülich Ozone Sonde

Intercomparison Experiment: calibration functions traceable to one ozone reference instrument. Atmospheric Measurement Techniques, 17, 73–112, https://doi.org/10.5194/amt-17-73-2024.

Thouret, V., Clark, H., Petzold, A., Nédélec, P., Zahn, A., 2022. IAGOS: Monitoring Atmospheric Composition for Air Quality and Climate by Passenger Aircraft. In Handbook of Air Quality and Climate Change (pp. 1-14). Singapore: Springer Nature Singapore.